# High-Throughput Sequencing Reveals Novel microRNAs Involved in the Continuous Flowering Trait of Longan (*Dimocarpus longan* Lour.)

**DOI:** 10.3390/ijms232415565

**Published:** 2022-12-08

**Authors:** Saquib Waheed, Fan Liang, Mengyuan Zhang, Dayi He, Lihui Zeng

**Affiliations:** Institute of Genetics and Breeding in Horticultural Plants, College of Horticulture, Fujian Agriculture and Forestry University, Fuzhou 350002, China

**Keywords:** longan, continuous flowering, microRNA, novel-miR137

## Abstract

A major determinant of fruit production in longan (*Dimocarpus longan* Lour.) is the difficulty of blossoming. In this study, high-throughput microRNA sequencing (miRNA-Seq) was carried out to compare differentially expressed miRNAs (DEmiRNAs) and their target genes between a continuous flowering cultivar ‘Sijimi’ (SJ), and a unique cultivar ‘Lidongben’ (LD), which blossoms only once in the season. Over the course of our study, 1662 known miRNAs and 235 novel miRNAs were identified and 13,334 genes were predicted to be the target of 1868 miRNAs. One conserved miRNA and 29 new novel miRNAs were identified as differently expressed; among them, 16 were upregulated and 14 were downregulated. Through the KEGG pathway and cluster analysis of DEmiRNA target genes, three critical regulatory pathways, plant–pathogen interaction, plant hormone signal transduction, and photosynthesis-antenna protein, were discovered to be strongly associated with the continuous flowering trait of the SJ. The integrated correlation analysis of DEmiRNAs and their target mRNAs revealed fourteen important flowering-related genes, including *COP1*-*like*, *Casein kinase II*, and *TCP20*. These fourteen flowering-related genes were targeted by five miRNAs, which were novel-miR137, novel-miR76, novel-miR101, novel-miR37, and csi-miR3954, suggesting these miRNAs might play vital regulatory roles in flower regulation in longan. Furthermore, novel-miR137 was cloned based on small RNA sequencing data analysis. The pSAK277-miR137 transgenic *Arabidopsis* plants showed delayed flowering phenotypes. This study provides new insight into molecular regulation mechanisms of longan flowering.

## 1. Introduction

*Dimocarpus longan* Lour, often called longan or dragon eye, is a tropical tree that bears nutritious fruit [1]. As a member of the soapberry family (Sapindaceae), it is one of the most well-known tropical and subtropical species. It grows satisfactorily in tropical or subtropical countries such as China, Thailand, Vietnam, India, and South Africa. However, it is commercially exploited only in China and Thailand [2].

It is generally recognized that flowering is a significant event in plant life, particularly in fruit trees. Flowering is an essential developmental process in the plant’s life cycle; regardless of the time of harvesting fruit, flowering at the optimum time is critical for fruit set and crop productivity [3]. Typically, most longan varieties have a seasonal flowering cycle, such as ‘Lidongben’ (LD) [4]. Certain conditions are required to induce floral bud development in longan, such as a period of low temperature (vernalization), adequate salinity, and a dry environment [5]. During the off-season, the flowering of longan is induced by chemical treatment, including potassium chlorate (KClO_3_) [6,7], and the induction impact is heavily influenced by region and tree species. However, the longan cultivar ‘Sijimi’ (SJ) originated in the border region between China (Guangxi Province) and Vietnam [8] and has a continual flowering trait due to a spontaneous mutation. It blooms and bears fruit throughout the year, in tropical and subtropical climates, without requiring special environmental conditions. Axillary and terminal buds of SJ can differentiate into inflorescences [5]. It is possible to observe both flowers and fruits on one tree simultaneously. Furthermore, when the SJ tree is headed back heavily, sprouting shoots can bloom once becoming mature. Thus, SJ is an excellent model for studying the continuous flowering of fruit trees.

MicroRNAs (miRNAs) play a vital role in regulating flowering by acting as post-transcriptional regulators [9]. MiRNAs are non-coding small RNAs (sRNA) approximately 18 to 24 nucleotides (nt) in length and can inhibit the translation of target genes [10]. They are highly conserved among species across the plant and animal kingdoms, yet no single miRNA has sequence similarity with another miRNA in either lineage [11]. Many miRNAs have been identified in plants with a crucial role during the developmental process. The flowering cycle is imperative for plant reproduction and evolution. To reproduce successfully, plants must undergo distinct phases throughout their lives, from juvenile to adult and adult to reproductive phase [12,13]. It is accomplished by modulating critical flowering-time genes’ transcriptional and post-transcriptional expression. MiRNAs associated with flowering time play a vital role in the development of plants during the vegetative to the reproductive phase transition [13].

Through their interaction with biochemical and environmental factors, miRNAs and their targets influence the timing of flowering and cross-talk with other miRNA pathways [3]. The importance of plant miRNAs in the floral transition is becoming increasingly evident. An abundance of research published in the last few years demonstrates the critical role of miRNAs in controlling the expression of the genes involved in flowering time, floral transition, and floral bud development. For instance, several miRNAs have been identified to function in controlling flowering. For example, the miR156, miR172, miR169, miR159, and miR399, as well as their target genes, play a pivotal role in flowering regulation [3,14,15,16]. By controlling the expression of an *AP2*-*like* gene, miR172 influenced flowering time [17], floral organ characteristics [18], and flower certainty [19]. Using an activation-tagging technique, it was determined that overexpression of miR172 in *Arabidopsis* induces early blooming and disrupts the determination of floral organ identity [17]. It was discovered that miR156 and miR157 suppress the translation of *SPL3* mRNA to prevent early blooming. The lowest levels of miR156 and miR157 expression were observed during adult leaf and inflorescence growth, enabling *SPL3* and other genes to accumulate and influence flower formation [15]. At least five distinct blooming pathways have been identified in *Arabidopsis* [20], each of which is regulated by a unique set of floral integrin genes, including *flowering locus T* (*FT*), *flowering locus C* (*FLC*), and *CONSTANS* (*CO*) [21].

However, the information related to the molecular mechanism of flowering in perennials is comparatively limited compared to model plants due to long generation times and complex genetic backgrounds. Specific genes associated with flowering time in *Arabidopsis* do not consistently influence blossoming in trees. For example, poplar overexpressing *MADS1* and *CO*, *LFY*, *AP1*, and *agamous-like20* (*AGL20*) resulted in very early flowering or no flowers in the transgenic lines, indicating that perennials have different mechanisms for regulating flowering [22].

Various environmental and biological factors affect longan fruit yield significantly, but the difficulty and instability of blossoming are the most significant concerns. The ability to flower at the right time of year is an imperative characteristic of fruit trees that directly impacts production. Studying the molecular mechanisms of floral induction in longan is essential for better understanding flowering-related problems. However, due to the long generation time, such knowledge regarding floral induction is rare in longan. The molecular mechanisms underlying the flowering traits of SJ remain unknown despite some RNA sequencing analyses [4,5,23]. In this study, miRNA-Seq analysis was performed to identify differentially expressed miRNAs (DEmiRNAs) associated with continuous flowering using two longan cultivars (SJ and LD). We aim to elucidate the genetic foundation for the different flowering characteristics observed in two longan cultivars during floral induction. The results of this study may provide valuable information regarding the molecular regulatory mechanisms of floral induction in two longan cultivars that differ in flowering time characteristics.

## 2. Results

### 2.1. Data Quality Analysis

We sequenced four sRNA libraries for this study. Total raw read counts for these sRNA libraries were 34,510,998 (SJ) and 33,791,363 (LD) (Table 1). After removing the junk sequences and low-quality reads, we obtained the remaining 30,483,736 clean reads from SJ (88.33%) and 30,976,435 clean reads from LD (91.67%), which were mapped to the reference genome to identify prospective candidate miRNAs (Table 1). In both samples, the fraction of clean reads exceeded fifty percent, indicating that the quality of the sequencing data was good. The correlation coefficient between SJ and LD was more than 0.75 (Figure 1), showing that the molecular components of the floral induction response largely overlap.

### 2.2. Prediction of Known and Novel miRNAs

The clean reads were searched against the miRBase (v21) database to identify conserved miRNAs. In total, 1662 known miRNAs belonging to different miRNA families were identified. There was significant variation in the number of members within miRNA families. The unmatched sRNA reads were aligned with the reference genome by using MiRDeep2 to identify novel miRNAs. A total of 235 novel miRNA candidates showed stable hairpin structures and were designated as novel_miR01 to novel_miR235. Most of the novel miRNAs ranged in length from 18 to 25 nt, with 24 nt lengths being the most abundant (Figure 2A).

An miRNA’s mature sequence and specific target are largely determined by its first cleavage position [24,25]. MiRNAs are composed of various bases, which contribute to their secondary structures and biological properties. In the 21-nt and 22-nt putative novel-miRNAs, the U nt was dominant as the first nt. The high percentage of 23-nt and 24-nt novel miRNAs begin with A, in contrast to 18-nt and 19-nt novel miRNA candidates that begin with C (Figure 2B). Further, the nt bias of each miRNA showed that novel miRNAs with A at the 5’ end were the most prevalent. Previous research indicated that the first nt is necessary for miRNA sorting [25] and that the tenth and eleventh nts are critical for directing the miRNA to cleave the target mRNA [26]. It was predominantly detected in novel miRNAs, with G being the most common base, followed by A and U (Figure 2C).

### 2.3. Small RNA Profiles and miRNA Identification

Potential target genes were inferred using the Target Finder tool to comprehend the identified miRNAs’ role further. The results demonstrated that 1868 of 1898 miRNAs possess target genes (Table 2). Approximately 13,334 genes were predicted to be targeted by different miRNAs. A total of 11,728 genes were annotated and predicted to be putative targets of various conserved miRNA families. The expected number of targets per miRNA ranged from 1 to 457. In certain miRNAs, a single gene was targeted by multiple miRNAs, consistent with previous observations on the crucial functions of miRNAs in plants for regulating diverse biological processes.

The critical miRNAs that exhibit different flowering phenotypes during floral induction in two longan cultivars were identified using a cluster analysis of expression patterns. A heat map displaying their unique expression profiles was established (Figure 3A). One conserved miRNA and 29 novel miRNAs were identified as differently expressed; 16 were upregulated, and 14 were downregulated (Figure 3B, Table 2). Five expression profiles characterized these miRNAs. Interestingly, except for csi-miR3954, all differently expressed miRNAs were novel miRNAs. Furthermore, several putative novel miRNAs showed distinct expression profiles with greater variation levels between the SJ and LD libraries.

### 2.4. Function and Pathway Analysis of DEmiRNA Target Genes

To clarify the biological processes/pathways and the interaction between two longan cultivars, the functional characterization of DEmiRNA target genes was carried out using GO term and KEGG pathway enrichment analysis. GO analyses revealed that the potential roles of 818 miRNA targets could be categorized into 11 cellular components, 13 molecular activities, and 17 biological processes (Figure 4A). Among them, the DEmiRNA target genes relevant to cellular components were 230. The three largest groups that contained the most genes were cell part (178), cell (178), and cell part (138). The DEmiRNA target gene relevant to molecular functions was 367. The catalytic activity (273) and binding (217) terms occupied the most significant numbers of genes. A total of 355 DEmiRNA target genes were relevant to biological processes. The three most important biological processes were the metabolic process (269), cellular process (248), and response to stimulus (76).

Furthermore, analysis of enriched KEGG pathways of the target genes of DEmiRNAs revealed that the plant–pathogen interaction pathway had the most targets (containing 21 genes), followed by the biosynthesis of amino acids pathway with nine genes and the signal transduction pathway with nine genes (Figure 4B). The pathway significant enrichment analysis can assist us in determining which DEmiRNA targets are implicated in particular pathways. Plant–pathogen interaction and the photosynthesis-antenna protein were the most highly expressed pathways in the upregulated groups (Appendix A). This suggests that genes related to these pathways are likely to play an important role in flower regulation in longan.

### 2.5. Correlation Analysis of Differentially Expressed miRNAs and Target mRNAs

Analyzing miRNA expression profiles linked to their predicted or validated targets can provide indirect evidence that miRNAs are involved in the cleavage or inhibition of target mRNAs. To develop a DEmiRNA-mRNA regulation network, the pairs of DEmiRNA and its target genes with negative correlations −0.7 of their expression levels were screened. As shown in Figure 5, 48 DEmiRNA-target pairs showed correlations that contained 13 DEmiRNAs. Several target genes were identified in the integrated correlation analysis, including *COP1*-*like*, *Casein kinase II*, and *TCP20*, as important flowering-related genes (Table 3). The identified flowering-related genes in correlation analysis were targeted by five DEmiRNAs, respectively. 

### 2.6. Validation of the miRNAs and Flowering-Associated Target Genes

Based on the regulatory pathways, hierarchical heat map DEmiRNAs, and their targets, five DEmiRNAs were selected for further validation through qRT-PCR. In general, the expression of these miRNAs was similar to sequencing data, with slight variation. It was found that novel-miR137 was significantly downregulated in the SJ compared to LD (Figure 6), whereas csi-miR3954, novel-miR76, novel-miR37, and novel-miR-101 were significantly upregulated in the SJ. The results of the qRT-PCR were almost consistent with sequencing data. It indicates that the results from sRNA sequencing are reliable and may provide an accurate indication of the level of miRNA expression in longan.

The expression profiles of 14 flowering-related targets were also validated by qRT-PCR. The results revealed that the expression pattern of genes varied between the SJ and LD and might play different roles during the floral transition in longan. The expression profiles of *Dof1.1*, *kinase WNK4*, *ABF2*, *FTIP1*, *SPA1-RELATED, RGLG2*, *COP1-like*, and *kinase TMK1-like*, regulated by novel-miR137, showed opposite expression profiles, as expected for miRNA targets (Table 3, Figure 6).

### 2.7. Overexpression of Dlo-Novel-miR137 Altered Phenotypes of Transgenic Arabidopsis Thaliana

As novel-miR137 is predicted to target the key flowering genes such as *Dof1.1*, *kinase WNK8*, and *RGLG2*, indicating its role in regulating the flowering process. The over-expression vectors for a novel miR137 were constructed and transformed into *Arabidopsis* (Appendix A). The transgenic plants were confirmed by PCR analysis (Appendix A). Following the segregation tests, no less than ten plants of three independent T3 homozygous lines were used for flowering time and phenotype analysis. It was found that *Arabidopsis* plants overexpressing dlo-novel-miR137 showed a delay in flowering time and an increase in the number of rosette leaves (Figure 7A). Under the same conditions, the average flowering time and rosette leaf number of WT *Arabidopsis* plants were 22.3 ± 0.5 days and 11.5 ± 0.4, respectively (Figure 7B,C). For transgenic line miR137-1, which was the most delayed, the average flowering time was 25.3 ± 0.4 days, followed by line miR137-2 (24.7 ± 0.4 days) and line miR137-4 (24.0 ± 0.5 days), respectively (Figure 7B). 

## 3. Discussion

The early-flowering/precocious feature benefits woody horticultural plants (fruit trees) by facilitating early fruit setting and timely harvest. A complex gene network controls the transition from vegetative growth to reproductive development by combining diverse environmental and endogenous cues in concert [26,27,28]. In various species, miRNAs and target genes have been associated with regulating flowering time [26,29,30]. Several recent RNA-Seq studies have examined flowering genes in SJ [4,5,23]. However, the underlying molecular mechanisms of continuous flowering in SJ remain unknown. To clarify the genetic basis for the floral transition of SJ, this study performed a miRNA sequencing analysis.

Most of the known miRNA families in longan have been found in other species, including *Arabidopsis thaliana* [31], *Brassica napus* [32], *Oryza sativa* [33], *Solanum tuberosum* [34], *Zea mays* [35], *Phaseolus vulgaris* [36], and *Brachypodium distachyon* [37]. It is estimated that 30% of the miRNA families investigated are present in at least ten distinct plant species. According to the current and previous studies [38], evolutionarily conserved miRNAs exhibited a larger number of sequences (sequencing frequency) than non-conserved miRNAs [38]. Using criteria for probable pre-miRNA stem-loop structure and the biogenesis of existing miRNAs, 235 novel miRNAs were predicted in addition to the prediction of known miRNAs [39]. The number of novel miRNAs was less than the total number of known miRNAs. In addition, the absolute sequencing frequency of novel miRNAs decreased significantly. This finding was in line with previous studies [40,41,42], which showed that most species-specific novel miRNAs had higher spatiotemporal expression and lower sequencing frequencies than their conserved counterparts.

DEmiRNAs and their targets were examined to monitor transcriptional changes between the two longan cultivars (SJ and LD). Among the 30 miRNAs that showed significant differential expression, 16 were upregulated, and 14 were downregulated in SJ compared with LD. Interestingly, 29 out of 30 DEmiRNAs identified were novel miRNAs. In the present study, we observed that novel miRNAs have significantly different expression patterns, suggesting that they may play more important roles in floral control in longan. According to earlier findings concerning novel miRNAs in *Arabidopsis* and other plant species [31,43], most of the predicted target genes for novel miRNAs identified in this study encode plant-specific transcription factors, including stress transcription factors, hormone signal transduction genes, and genes associated with flowering pathways. Further investigations revealed that most of these transcription factors belong to the regulatory pathways associated with the plant–pathogen interaction, the signal transduction pathway of plant hormones, and the photosynthesis-antenna protein. Among the upregulated pathways, plant–pathogen interaction and photosynthesis-antenna protein played significant roles, indicating that related genes and transcription factors play an instrumental role in regulating the longan flower.

Advanced plant research has resulted in the identification of hundreds of transcription factors [44,45]. They are crucial for the development of plants’ morphology as well as their ability to withstand environmental stress [46,47]. Winterhagen et al. [48] demonstrated that chlorite and hypochlorite might directly trigger a stress response, raise cytokinin levels, and stimulate gene expression in longan flowering. In the present study, we found various transcription factors, such as *WRKY*, *HSP20*, *Dof*, and *MADS*-*Box*, participate in plant–pathogen interaction and photosynthesis-antenna protein pathways.

We found that novel-miR137 targeted the *COP1* gene, and the pathways showed that COP1 protein represses the expression of *GIGANTEA* (*GI*) in the circadian clock-controlled flowering pathway. *GI*, an essential component of the circadian clock-driven flowering pathway, controls *CO* transcription under inductive light settings [49,50]. *GI* is a target of miR172 in *Arabidopsis*, and the expression level of miR172 was significantly reduced in *GI* mutants (*gi*-*2*) [50]. Long-day conditions showed a significant increase in miR172 abundance compared to short-day conditions, both in wild-type and *gi*-*2* mutants. Moreover, the SUPPRESSOR OF PHYA-105 (SPA) protein is another critical flowering time gene, being a member of a small four-member family that is required for normal elongation and suppression of photomorphogenesis in adult plants [51,52]. A normal photoperiodic flowering is dependent upon *SPA1* among the four *SPA* genes. It has been demonstrated that mutations in the *SPA1* gene cause early flowering under short-day conditions [53,54]. Moreover, mutations in *SPA1* further disrupted the flowering process in short-day to the point where the flowering time no longer depended on the day length [55]. *SPA1* is the primary factor in controlling flowering time. SPA proteins have been shown to interact with another repressor of light signaling, the ubiquitin ligase COP1, to ubiquitinate light signaling activators in dark-grown seedlings [56]. It is postulated that SPA proteins interact with COP1 to ubiquitinate light response activators [57]. In this work, qRT-PCR indicated that *SPA1*, *COP1*, and *FTIP1* expression levels were elevated in SJ and downregulated in LD. It is likely that *SPA*/*CO*/*FTIP1* functions together and interacts with other flowering time-related genes, contributing to the continuous flowering trait of SJ. Nonetheless, further research is required to comprehend the processes underlying the interaction between *SPA*/*CO*/*FTIP1* function.

Furthermore, the presence of *CK2* is essential for regulating circadian rhythms, hormone responses, light signaling, and flowering time control [57,58]. The CK2 protein has an evolutionarily conserved role as a component of the circadian clock in a variety of organisms, including diverse plant species, wheat, rice, tobacco, maize, mustard, and *Arabidopsis* [55,59,60]. In this study, the *CK2* gene was DEmRNA in the integrated analysis of the transcriptome. According to the miRNA sequencing data predicted to be targeted by novel-miR37 in the circadian rhythm–plant pathway. According to the qRT-PCR results, *CK2* levels were decreased in SJ and increased in LD. There has been evidence from previous studies that show that plants with triple mutants (*CK2 α1α2α3*) exhibit late flowering phenotypes under both long-day and short-day conditions [61,62]. Flowering time is modulated by genes that encode the *CK2* subunit, but the mechanism is not entirely understood. Recent studies have shown that miR397b regulates the flowering process by targeting *CK2*, which modulates the circadian period of the *CIRCADIAN CLOCK ASSOCIATED1* (*CCA1*) gene [63,64]. A miR397b-*CKB3*-*CCA1* circadian regulation feedback circuit was formed when *CCA1* binds directly to the promoter of MIR397B and suppresses its expression. *CK2* controls the stability and activity of both positively and negatively acting TFs in light signaling pathways. Consequently, these genes might play a crucial role in the emergence of the floral meristem to promote perpetual flowering traits.

Plant hormones affect many aspects of the plant’s life cycle, including flower development, stress responses, and secondary metabolites [65,66]. It was predicted that novel miR137 targeted two transcription factors related to signal transduction in the plant hormone signal transduction pathway, the *GH3* auxin-responsive transcription factor and *histidine kinase 3* (*HK3*). *GH3*-overexpressing mutants, such as *dfl1*-*D* and *dfl2*-*D* [67], display reduced growth and altered light responses, which raises the possibility that these GH3 proteins are involved in light and auxin interactions. In recent years, biochemical studies have been conducted on the GH3 proteins. Auxin may be responsible for regulating auxin homeostasis by rapidly inducing the expression of *GH3* genes [68,69]. Since auxin only partially induces the *GH3* genes, their functional processes might not be as simple. An important role of auxin is in plant development and growth, including the induction of floral development [70]. According to the results of the present study, novel-miR137 specifically targets the *GH3* auxin-responsive promoter, which is down-regulated in SJ in accordance with the plant hormone signal transduction pathway. According to a recent study on the perpetual flowering trait of roses, expression levels of *GH3*, which is responsible for maintaining auxin homeostasis and converting auxin into amino acids, increase during floral induction in seasonal flowering roses. In contrast, expression levels were decreased in perpetual flowering roses [71]. The expression of *GH3* is down-regulated in SJ; hence, it may contribute to developing features associated with perpetual flowering by influencing the emergence of the floral meristem. *TMK1*, another gene involved in the auxin signaling system, was DEmRNA comparing the two longan cultivars. In *Arabidopsis*, the *TMK* subfamily of *Receptor*-*Like Kinases* plays a critical role in growth and exhibits decreased auxin sensitivity [72]. The biological function of *TMK1* has not yet been fully determined despite these molecular and biochemical analyses.

We report the isolation and characterization of novel-miR137 from longan. Our research observed the phenotype of delaying the flowering time in pSAK277-novel miR137 transgenic plants, suggesting that novel-miR137 might be a flowering repressor. We predicted that novel miR137 targets nine genes in longan, including *COP1*, *kinase TMK1-like*, *Dof1.1*, *SPA1-RELATED 3*, *kinase WNK4*, *RGLG2*, *FTIP1*, *ABF2* and *GH3*, which have been shown to play a critical role in flower regulation of various plants [45,64,73,74,75,76]. The expression level of novel miR137 is lower in SJ than in LD, and the expression levels of these target genes increase in SJ, which is confirmed by qPCR. Therefore, these results indicate that the lower expression of novel miR137 may promote the trait of continuous flowering of SJ longan by up-regulating its targeted flowering genes.

## 4. Materials and Methods

### 4.1. Data Retrieval and Plant Materials

Longan samples of SJ and LD were collected from the experimental fields of Fujian Agriculture and Forestry University in Fuzhou. Similar mature trees of two cultivars were selected and headed back heavily. The terminal tips of newly sprouting shoots of two cultivars (SJ and LD) were collected before maturity as plant materials. As the SJ cultivar has the continuous flowering trait, it is difficult to obtain shoots with the same developing phase from mature trees of SJ and normal longan cultivars. Therefore, the terminal tips of newly sprouting shoots after the heading back of mature trees were used as plant samples in this study to minimize the effect of inflorescence differentiation of SJ in mature trees. Two biological replicates per cultivar were used to reduce the variation in the expression levels of miRNAs between different trees, and statistical tests were applied to determine whether there were significant differences. In the present study, the SJ1 and SJ2 represent SJ and LD1, and LD2 represents LD. After harvest, all samples were immediately frozen in liquid nitrogen and stored at −80 °C for later analysis. The RNA-seq data of the two cultivars (SJ and LD) were obtained from our previous report [4]. The *Arabidopsis* WT Col-0 and transgenic *Arabidopsis* were grown at 23 ± 2 °C and 75% relative humidity on long days (16 h light/8 h dark).

### 4.2. Total RNA Extraction and Library Construction

Total RNA was isolated using TRIzol reagent (Life Technologies, Carlsbad, CA, USA) and treated with RNase-free DNase I (Takara Biotechnology, Beijing, China) according to the manufacturer’s instructions. An ND1000 spectrophotometer was used to quantify and check all RNA samples for protein contamination (A260/280) and reagent contamination (A260/230). As a starting amount, 1.5 µg of RNA was used, 6 µL of water was added to the RNA samples, and libraries were constructed. The libraries of sRNA fragments sized between 18 and 30 nt were constructed from the sRNAs separated by gel separation and screened using gel extraction. Qubit 2.0 was used to test the library concentration. The library concentration was diluted to 1 ng/µL, and the Insert Size was determined using an Agilent 2100 bioanalyzer. To ensure the quality of the library, the q-PCR method was utilized to quantify the effective concentration of the library accurately. Following the qualification of the library, high-throughput sequencing was performed. Sequencing was conducted on the Illumina HiSeq X Ten platform, and sequencing read lengths were single-end (SE) 50 nt.

### 4.3. Identification of Known and Novel miRNAs

The raw deep-sequencing data were preprocessed to eliminate low-quality tags, yielding sRNA tags. The Cutadapt toolkit was utilized to remove low-quality reads, sequences with a poly-A tail, adapter sequences, and reads with <18 or >30 bases [77]. In addition, the clean sRNA reads were then aligned to the GenBank and Rfam 12.2 (http://rfam.xfam.org/, accessed on 19 January 2020) databases using BLAST searches and bowtie to screen and remove sequences associated with other types of small RNAs (rRNA, scRNA, snoRNA, snRNA, and tRNA). The sequence alignment and subsequent analyses were carried out using the *Dimocarpus longan* genome as a reference (ftp:/climb.genomics.cn/pub/10.5549 101000/100276/, accessed on 19 January 2020). To identify known miRNAs, we matched the reference genome’s read sequences with mature miRNA sequences from the miRNA database miRBase (v21). Reads having identical sequences to reported miRNAs were regarded as known miRNAs. The final miRNA dataset analyzed each site’s sequence length distribution and nt preference. Meanwhile, miRDeep2 [78] was used to investigate the remaining unannotated sRNA sequences that were not matched to any pre-miRNAs in miRbase. We utilized the miRDeep2 software program to obtain probable precursor sequences by matching reads to genome position information based on read distribution (mature, star, loop) and precursor structure energy. A Bayesian model scoring (RNAfold randfold) to predict novel miRNAs. Levels of miRNA expression were calculated using the transcripts per kilobase million (TPM) method. To quantify the similarity between the two variables, we employed the Pearson correlation coefficient, also known as the Pearson product-moment correlation coefficient.

### 4.4. Bioinformatics Analysis of Differentially Expressed miRNAs

We utilized DEGseq [70] to determine DEmiRNAs. The DEmiRNA was identified based on the following parameters: |log2(FC)| ≥ 1; FDR ≤ 0.01. Since differential miRNA expression analysis is an independent statistical hypothesis test, there is a probability of false positives. The Benjamini-Hochberg technique is used to assess the original hypothesis test’s significance p. False discovery rate (FDR) was used to filter DEmiRNAs after the *p*-value was adjusted.

### 4.5. Prediction of miRNA Targets and Enrichment Analyses

Target Finder [79] was used with default parameters to predict the miRNA target genes. BLAST was used to match target gene sequences with the NR [80], Swiss-Prot [81], GO [82], COG [83], KEGG [84], KOG [85], and Pfam [86] databases. Enrichment analysis of different genes between sample groups was carried out using the topGO program. The annotation and enrichment analysis of biochemical pathways were conducted using the Kyoto Encyclopedia of Genes and Genomes (KEGG), and Fisher’s exact test was used to determine the significance of the enrichment.

### 4.6. Validation of miRNA and Target Gene Expression via QRT-PCR Analysis

qRT-PCR was used to assess longan miRNA expression levels. Reverse transcription was performed using a Mir-XTM miRNA First-Strand Synthesis Kit (Takara, Dalian, China). qRT-PCR was performed on an Applied Biosystems 7500 Real-Time PCR System using a Takara Mir-XTM miRNA qRT-PCR TB Green^®^ Kit. The forward primers used for qRT-PCR amplification of each miRNA are listed in Appendix A, and the reverse primer was U6 from the kit. The thermal cycling conditions were an initial polymerase activation step for 30 s at 95 °C followed by 40 cycles of 5 s at 95 °C for template denaturation and 34 s at 60 °C for annealing. Stage 3 was 15 s at 95 °C, 1 min at 60 °C, and 15 s at 95 °C. Using the 2^−ΔΔCT^ approach, the expression levels were quantified using the raw fluorescence data from the 7500 Real-Time PCR detection equipment.

### 4.7. Vector Construction and Plant Transformation

To verify the impact of dlo-novel-miR137 on flowering, we constructed a vector for overexpression of dlo-novel-miR137. The primer pair of MIR137-F (5′-GGCATACGAGACTGAGGCTC-3′) and MIR137-R (5′-CGTGTCTGACTGAGTGCTGT-3′) was used to amplify a 645-bp fragment from the longan DNA. The precursor sequence of dlo-novel-miR137 were fused into the plasmid of pSAK277 and then introduced into *Agrobacterium tumefaciens* strain GV3101 for transformation. *Agrobacterium tumefaciens* strain GV3101 containing the recombinant expression vector pSAK277-MIR137 was cultured at 28°C overnight, cultures were harvested and resuspended in the infiltration buffer to a final OD600 = 0.8, and transformed into *Arabidopsis* by the floral dip trans-formation method [87]. Transgenic lines were cultured on half-strength Murashige and Skoog (MS) medium supplemented with 100 mg/L kanamycin. After 15 days, the plants with normal, healthy, green cotyledons were transplanted into pots filled with artificial soil. The phenotypes of transgenic plants up to their third generation were analyzed after being subjected to kanamycin selection. At least 10 separate transgenic plants with significant traits were selected and investigated for phenotypic characterization.

## Figures and Tables

**Figure 1 ijms-23-15565-f001:**
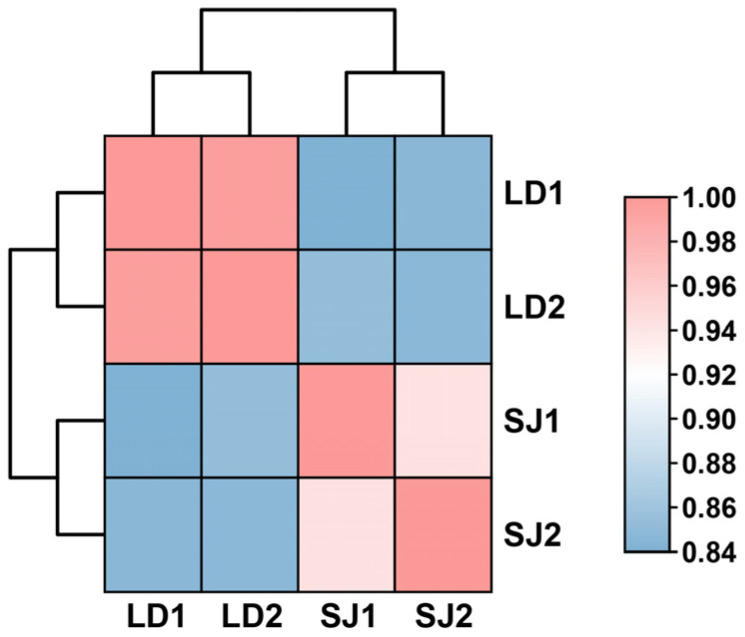
The correlation analysis between four samples based on FPKM results. Different colors in the figure represent different correlation coefficient values. The horizontal and vertical coordinates represent different samples.

**Figure 2 ijms-23-15565-f002:**
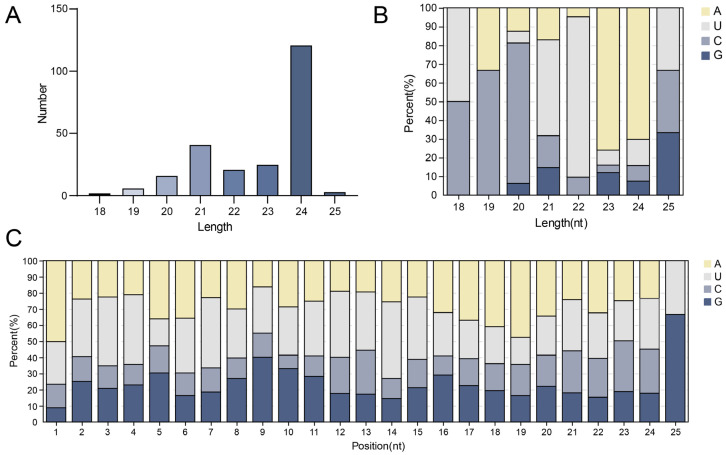
Predicted novel miRNAs length profile and proportion of nucleotide bias at each position within novel-miRNAs in the library of the small RNAs of longan. (**A**) Length profile of novel miRNAs. (**B**) The proportion of first nucleotide bias. (**C**) The proportion of nucleotide bias at each position.

**Figure 3 ijms-23-15565-f003:**
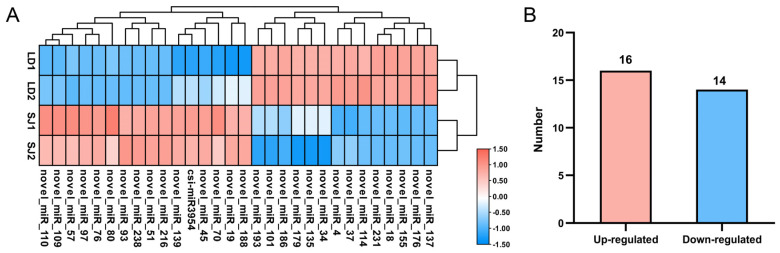
Differential expression analysis of miRNAs. (**A**) Heat map of longan differentially expressed miRNAs in ‘SJ’ and ‘LD’. The scale bar corresponds to miRNA relative expression levels; different hues indicate relative expression levels. Pink and blue correspond to up− or down−regulated expression of a given miRNA. (**B**) The number of up−regulated and down−regulated DEmiRNAs.

**Figure 4 ijms-23-15565-f004:**
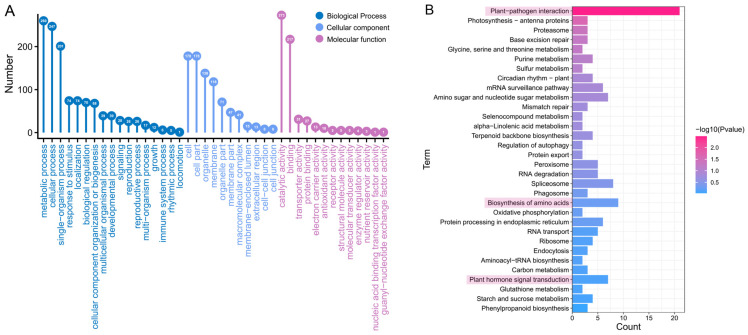
The GO and KEGG classification of differentially expressed miRNA target genes. (**A**) GO annotation classification statistics. (**B**) KEGG enrichment analysis.

**Figure 5 ijms-23-15565-f005:**
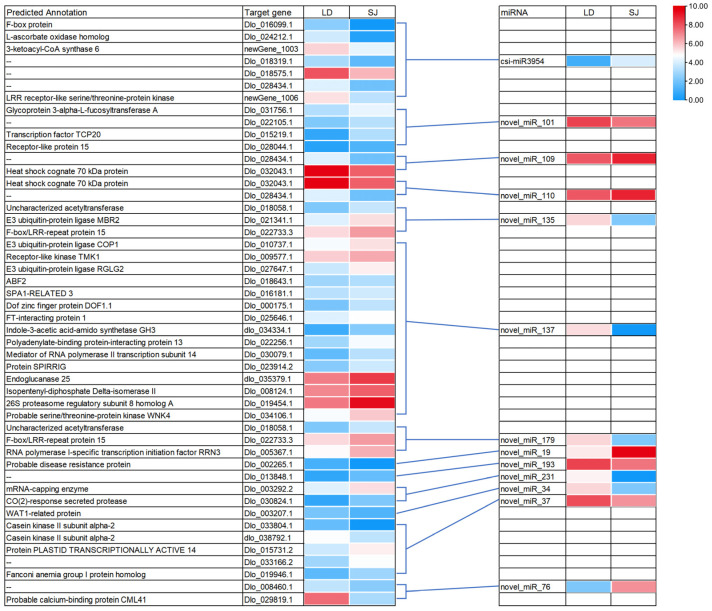
A combined view of correlation expressions between differentially expressed miRNAs and their target compression in ‘SJ’ vs. ‘LD’.

**Figure 6 ijms-23-15565-f006:**
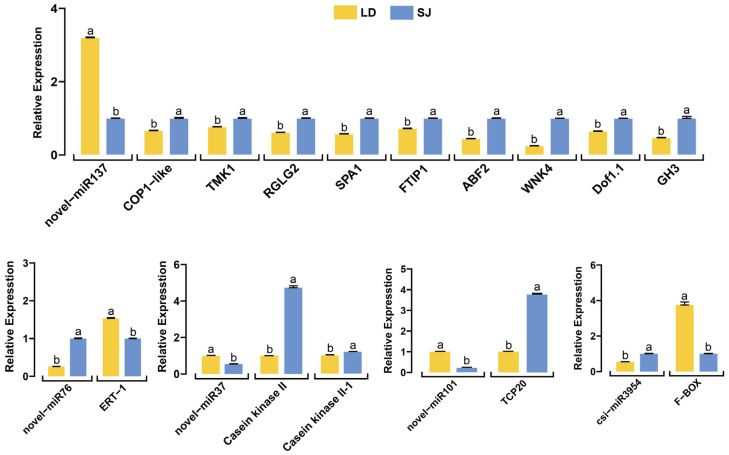
Relative expression levels of five miRNAs and their target genes. Validated by qRT-PCR analysis and calculations were carried out using the 2^−ΔΔCT^ method. The vertical bars represent each mean value’s ± SE (standard error). The different letters (a, b) show significant differences at the *p* < 0.05 level.

**Figure 7 ijms-23-15565-f007:**
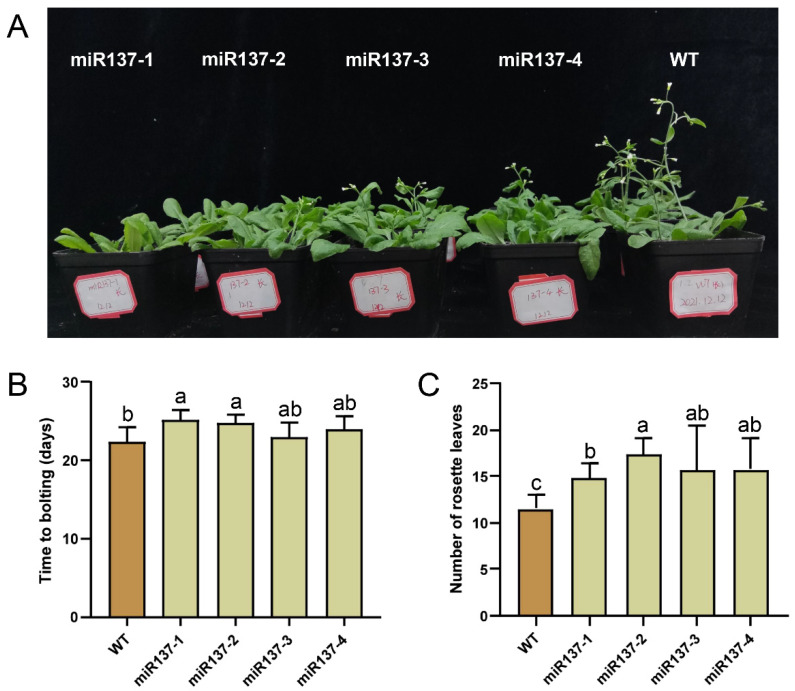
Phenotypic analysis of transgenic *Arabidopsis* plants overexpressing dlo-novel-miR137. (**A**) Flowering phenotype at 25 days under long photoperiod. The miR137-1, miR137-2, miR137-3 and miR137-4 represented four transgenic *Arabidopsis* lines, respectively. (**B**) Flowering time of bolting. (**C**) The number of rosette leaves at flowering. The different letters (a, b, c) show significant differences at the *p* < 0.05 level.

**Table 1 ijms-23-15565-t001:** Sequencing data statistics output.

Sample	Raw Reads	Clean Reads	Q30 (%)
SJ1	16,420,921	14,098,627	98.69
SJ2	18,090,077	16,385,109	98.95
LD1	15,250,206	13,886,485	98.65
LD2	18,541,157	17,089,950	98.85

Note: SJ1, SJ2 represents ‘Sijimi’, and LD1, LD2 represents ‘Lidongben’.

**Table 2 ijms-23-15565-t002:** Predicted known and novel miRNAs with targets.

Types	All miRNA	miRNA with Target	Target Gene
Known miRNA	1662	1659	10,827
Novel miRNA	235	209	4252
Total	1897	1868	13,334

**Table 3 ijms-23-15565-t003:** Prediction of differentially expressed miRNA targets related to floral induction.

MiRNA	Target Gene	Name	Functional Annotation
novel-miR137	Dlo_010737.1	COP1-like	Photoperiodism, flowering (GO:0048573); entrainment of the circadian clock (GO:0009649)
	Dlo_009577.1	kinase TMK1-like	Auxin signal transduction and activation of MAPKK activity (GO:0000186)
	Dlo_027647.1	RGLG2	Intracellular auxin and metal ion binding (GO:0046872)
	Dlo_016181.1	SPA1-RELATED 3	Protein kinase activity (GO:0004672)
	Dlo_025646.1	FTIP1	FT-interacting protein 1
	Dlo_018643.1	ABF2	Transcription factor binding
	Dlo_034106.1	kinase WNK4	Vegetative to the reproductive phase transition of the meristem (GO:0010228)
	Dlo_000175.1	Dof1.1	TF Dof domain, zinc finger
	Dlo_034334.1	GH3	indole-3-acetic acid amido synthetase activity (GO:0010279)
novel-miR76	Dlo_018664.1	ERT-1	Response to ethylene (GO:0009723); response to abscisic acid (GO:0009737)
novel-miR101	Dlo_015219.1	TCP20	Transcription factor TCP20
novel-miR37	Dlo_033804.1	Casein kinase II	Circadian rhythm (GO:0007623)
	dlo_038792.1	Casein kinase II	Circadian rhythm (GO:0007623)
csi-miR3954	Dlo_016099.1	F-BOX	F-box protein PP2-B2

## Data Availability

The datasets generated during and/or analysed during the current study are available from the corresponding author on reasonable request.

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
