# Peer review of "High-Throughput Sequencing Reveals Novel microRNAs Involved in the Continuous Flowering Trait of Longan (Dimocarpus longan Lour.)"

_ijms, 2022, doi:10.3390/ijms232415565_

Round 1
Reviewer 1 Report
1. Why you selected this two plant only?
2. What is novelty of this experiment?
3. methodolgy is not well described ?
4. Conclusion and Result is not matching ?
5. Result clearly show that WT is more better than other transgenic, why?
6.What is meaning of miR137-1, miR137-2, miR137-3 and miR137-4? which is not clearly mentioned in manuscript.
7. Less reference in disscusion.
Author Response
Point 1: Why you selected this two plant only?
Response 1: ‘Sijimi’ longan has a continuous flowering trait and is an excellent model for studying the flowering regulation of fruit trees. The ‘Lidongben’ longan is a typical longan cultivar, which has a seasonal flowering cycle, and used as the control. The purpose of our selection of these two plants is to elucidate the genetic foundation of the continuous flowering characteristics.
Point 2: What is novelty of this experiment?
Response 2: Five key miRNAs, including novel-miR137, novel-miR76, novel-miR101, novel-miR37, and csi-miR3954, were found to may play vital regulatory roles in flower regulation in longan. Novel-miR137 is isolation and characterization, transgenic Arabidopsis plants showed delayed flowering phenotypes suggesting it is a flowering repressor.
Point 3: Methodolgy is not well described ?
Response 3: We have modified the materials and methods, which can be seen in the revised manuscript. See Line 406-436 of the revised version.
Point 4: Conclusion and Result is not matching ?
Response 4: Thank you for your suggestion, we have improved the conclusion and corrected it according to the crucial results obtained in our study. see in the Line 365-373 of the revised version.
Point 5: Result clearly show that WT is more better than other transgenic, why?
Response 5: The over-expression vectors for a novel miR137 was constructed and transformed into Arabidopsis. It was found that Arabidopsis plants overexpressing Dl-novel-miR137 showed a delay in flowering time and an increase in the number of rosette leaves (Figure 7A). Under the same conditions, the average flowering time and rosette leaf number of WT Arabidopsis plants were 22.3 ± 0.5 days and 11.5 ± 0.4, respectively (Figure 7B 7C). While, transgenic Line miR137-1, which was the most delayed, the average flowering time was 25.3 ± 0.4 days, followed by line miR137-2(24.7 ± 0.4 days) and line miR137-4 (24.0 ± 0.5 days), respectively.
Because novel miR137 is a flowering repressor, so it repressed the flowering of the transgenic plants and the transgenic plants showed the delayed flowering phenotypes compared to WT.
Point 6: What is meaning of miR137-1, miR137-2, miR137-3 and miR137-4? which is not clearly mentioned in manuscript.
Response 6: We have modified the problem, which can be seen in the revised manuscript. see in the Line 245-246 of the revised version.
Point 7: Less reference in disscusion.
Response 7: Thank you for your suggestion, we have added more relevant references in the discussion part.
Reviewer 2 Report
In China, the longan tree (Dimocarpus longan Lour.) is a significant fruit tree. The difficulty of flowering is one of the most severe problems impacting D. longan fruit yield. Therefore, it is imperative to comprehend the molecular mechanisms that control longan blooming. The writing in the manuscript is excellent; it is precise, understandable, and well-written. The introduction section has all the necessary references and fully supports the study's goals. The study required a tremendous amount of labour, and from what I can see, the work is sound. The longan industry benefits from the study's conclusions, which are highly intriguing and original. The results are presented well, notably the gorgeous colour photos. According to the current study's findings, the novel miR137 in transgenic Arabidopsis delayed flowering time by controlling the expression of key flowering genes like WRKY, Dof2.4, COP1, FT-protein, and others. The discussion is very well written, supports the study's conclusion, and skillfully connects its findings to earlier research. As a result, I think the Editor-in-Chief should think about publishing this manuscript in this journal since it is a highly intriguing study.
Author Response
Thank the reviewers for their recognition of our research. We have improved the English writing.
Reviewer 3 Report
I have carefully read the manuscript entitled " High-throughput sequencing reveals novel microRNAs involved in the continuous flowering trait of ‘Sijimi’ longan (Dimocarpus longan Lour.)” which was submitted for consideration in the International Journal of Molecular Sciences (MDPI).
Dimocarpus longan is one of the better-known tropical tree species of the soapberry family Sapindaceae that produces edible fruit. This species is native to tropical Asia and China, but some authors believed it originate from the mountain range between Myanmar and southern China. Other reported origins include India, Sri Lanka, upper Myanmar, north Thailand, Kampuchea (syn. Cambodia), north Vietnam and New Guinea. Presently, D. longan is grown in southern China, Taiwan, northern Thailand, Malaysia, Indonesia, Cambodia, Laos, Vietnam, India, Sri Lanka, Philippines, Australia, the United States, Mauritius and Bangladesh. In this study, the authors analyzed the genetic foundation for the different flowering characteristics in two selected cultivars of this species. The authors’ results may provide valuable information regarding the molecular regulatory mechanisms of floral induction in studied taxa that differ in flowering time characteristics.
This manuscript is in general well written, logically structured, well-illustrated and easy to understand. It also addresses a subject of great interest in the scientific community. The title clearly describes the content of the article, although I propose to change it slightly (I have provided details below). The abstract is well written. The introduction is generally well written as it gives a good background of the research in question. However, I believe that this chapter should be supplemented with data on the research object, please indicate where the species comes from and what is its natural range. I believe that the Materials and Methods section is well-structured and scientifically sound. The results are well presented, figures and tables are correct. Literature reviews in the discussion section of the manuscript are very professional. My comments mostly relate to relatively minor issues of interpretation and writing. These comments do not influence a positive impression of the article.
Suggestions:
Title: Please consider changing the title to "High-throughput sequencing reveals novel microRNAs involved in the continuous flowering trait of Dimocarpus longan (Sapindaceae)"
Line 31-33: ” As a member of the soapberry family (Sapindaceae), it is one of the most well-known tropical species. There are several tropical and subtropical countries where it grows satisfactorily." - please complete this section, and provide the range of the species and detailed information on the geographical regions where it has been found.
Line 74 and throughout the text: Arabidopsis is the Latin name of the genus, I suggest that the taxon name be written in italics throughout the manuscript
Line 112: delete double space between “induction response”
Line 188-189: ” It suggests that genes related to these pathways are likely to play an important role in flower regulation in longan." - this is a suggestion, the sentence should be transferred to the discussion section
Line 255: Replace “brassica napus” with “Brassica napus”
Author Response
Point 1: Please consider changing the title to "High-throughput sequencing reveals novel microRNAs involved in the continuous flowering trait of Dimocarpus longan (Sapindaceae)".
Response 6: Thanks for your suggestions, we have revised the title, which can be seen in the revised manuscript (the following questions are revised here). See Line 3 of the revised version.
Point 2: Line 31-33: ” As a member of the soapberry family (Sapindaceae), it is one of the most well-known tropical species. There are several tropical and subtropical countries where it grows satisfactorily." - please complete this section, and provide the range of the species and detailed information on the geographical regions where it has been found.
Response 6: Thanks to the reviewer's suggestion, We added detailed information on the range of longan species and the geographic area where it has been found. It can be seen in the Line 32-33 of the revised version.
Point 3: Line 74 and throughout the text: Arabidopsis is the Latin name of the genus, I suggest that the taxon name be written in italics throughout the manuscript.
Response 6: Thanks to the reviewers for your suggestions. We have carefully checked the whole content, and similar errors have been corrected.
Point 4: Line 112: delete double space between “induction response”.
Response 6: Thanks to the reviewers for your suggestions. We have made changes in this part. See in Line 114 of the revised version.
Point 5: Line 188-189: ” It suggests that genes related to these pathways are likely to play an important role in flower regulation in longan." - this is a suggestion, the sentence should be transferred to the discussion section.
Response 6: Thanks to the reviewers for your suggestions. We have made changes in this part.
Point 6: Line 255: Replace “brassica napus” with “Brassica napus”.
Response 6: Thanks to the reviewers for your suggestions. We have made changes in this part. See in the Line 259.
Round 2
Reviewer 1 Report
Dear author,
The research paper is more exciting but some minor mistakes have been found. So, these mistakes will be correct.
With Regard

Author Response
Reviewer #1:
Point 1: line 298, Arabidopsis should be italicized.
Response 1: Thanks to the reviewers for your suggestions. We have made changes in this part. See in line 295 of the revised version.
Point 2: In 295, COP1 rectified it but the rest of the lines does-not correct it like wish- 294, 311, 312, etc. A similar kind of mistake does not correct GH3 in lines 349, 352, 355, 358, etc.
Response 2: Thanks to the reviewers for your suggestions. We have made changes in this part. We have carefully checked whether the full text gene names are italicized. The protein names are all changed without italics. See in line 291-310, 317, 341 of the revised version.
Point 3: In line 360, DEmRNA must be replaced with DEmiRNA.
Response 3: Thanks to the reviewers for your suggestions. DEmRNA is used to describe the TMK1 gene, so there is no error in the description here.
Point 4: 1. In the material and method section, 4.3. identification of known and Novel MiRNA should be corrected it. (Red color). 2. In line 423, Dimocarpus longan should be italicized. 3. A similar kind of mistake was in subheading 4.6. (MiRNA).
Response 4: We have made changes for such case errors. See in line 120, 144, 194, 208, 238, 397, 405, 419, 426, 434 of the revised version.